# Prematurity and Pulmonary Vein Stenosis: The Role of Parenchymal Lung Disease and Pulmonary Vascular Disease

**DOI:** 10.3390/children9050713

**Published:** 2022-05-12

**Authors:** Shilpa Vyas-Read, Nidhy P. Varghese, Divya Suthar, Carl Backes, Satyan Lakshminrusimha, Christopher J. Petit, Philip T. Levy

**Affiliations:** 1Department of Pediatrics, Emory University and Children’s Healthcare of Atlanta, Atlanta, GA 30322, USA; vyasre@emory.edu (S.V.-R.); suthard@kidsheart.com (D.S.); 2Department of Pediatrics, Pulmonology, Baylor College of Medicine, Houston, TX 77030, USA; npvarghe@texaschildrens.org; 3Department of Pediatrics, The Ohio State University Wexner Medical Center, Columbus, OH 43210, USA; carl.backes@nationwidechildrens.org; 4The Heart Center, Nationwide Children’s Hospital, Columbus, OH 43205, USA; 5Center for Perinatal Research, The Abigail Wexner Research Institute at Nationwide Children’s Hospital, Columbus, OH 43215, USA; 6Department of Pediatrics, UC Davis, Sacramento, CA 95616, USA; slakshmi@ucdavis.edu; 7Division of Pediatric Cardiology, Vagelos College of Physicians and Surgeons, Columbia University, New York Presbyterian Hospital, New York, NY 10065, USA; cjp2196@cumc.columbia.edu; 8Department of Pediatrics, Division of Newborn Medicine, Boston Children’s Hospital, Boston, MA 02115, USA

**Keywords:** pulmonary vein stenosis, prematurity, bronchopulmonary dysplasia, pulmonary vascular development, pulmonary hypertension, inflammation

## Abstract

Pulmonary vein stenosis (PVS) has emerged as a critical problem in premature infants with persistent respiratory diseases, particularly bronchopulmonary dysplasia (BPD). As a parenchymal lung disease, BPD also influences vascular development with associated pulmonary hypertension recognized as an important comorbidity of both BPD and PVS. PVS is commonly detected later in infancy, suggesting additional postnatal factors that contribute to disease development, progression, and severity. The same processes that result in BPD, some of which are inflammatory-mediated, may also contribute to the postnatal development of PVS. Although both PVS and BPD are recognized as diseases of inflammation, the link between them is less well-described. In this review, we explore the relationship between parenchymal lung diseases, BPD, and PVS, with a specific focus on the epidemiology, clinical presentation, risk factors, and plausible biological mechanisms in premature infants. We offer an algorithm for early detection and prevention and provide suggestions for research priorities.

## 1. Introduction

Pulmonary vein stenosis (PVS) is a heterogeneous disease process that contributes to morbidity and mortality in infants and young children [1,2]. PVS describes the pathologic process of the intraluminal narrowing of the veins that carry oxygenated blood from the lungs back to the left side of the heart [3]. Defined as a rare disease, PVS is an increasingly recognized complication of premature birth (<37 weeks) comprising close to 40% of all infant and childhood PVS [2]. PVS can be present after the intervention for pulmonary vein anomalies (e.g., total anomalous pulmonary venous connection) or following the repair of congenial heart disease in neonates (acquired PVS) [3,4,5,6]. Primary PVS in the absence of co-existing congenital heart disease, preceding cardiac surgical or catheter-based intervention, is an important comorbidity of chronic lung diseases of prematurity and can occur in up to 30% of patients with bronchopulmonary dysplasia (BPD) [2,7]. As a morphologic disruption of all components of the developing lung, including airway, vascular, and lymphatics features, BPD is characterized by the development of simplified alveolar structures, pathological vessel growth, and remodeling in the pulmonary arterial and venous beds. While several approaches exist to define BPD in preterm infants, all focus on the need for prolonged oxygen therapy and/or respiratory support [8]. In premature populations, primary PVS is detected later in infancy, suggesting additional overlapping postnatal factors with BPD disease development, progression, and severity [2,9].

There is also an established relationship in extreme premature infants between BPD and the development of pulmonary vascular disease (PVD), and its most severe form, pulmonary hypertension (PH) [7,10]. It has been suggested that mechanisms that result in BPD and chronic PH may contribute to the postnatal development of PVS [2,3,6]. The PVS that appears at several months of age in premature infants with cyanosis and dyspnea was initially thought to be a congenital abnormality of pulmonary vein formation [7]. However, the absence of lesions on initial echocardiography and the increasing severity and pulmonary vein involvement over time suggest that it may be gradual after birth, with both fetal programming and postnatal factors contributing to its pathogenesis and progression [2,6,11]. Premature infants with PVS have varying hemodynamic signatures during the neonatal course characterized by a degree of pulmonary venous congestion, interstitial edema, an upstream elevation of pulmonary artery pressure (PAP), and cardiac dysfunction. Many of the known dose-dependent risk factors that predispose to the development of BPD and PH (e.g., prolonged mechanical ventilation, oxygenation, inflammation, and exposure to a hemodynamically significant ductus arteriosus) may also contribute to PVS [12,13]. A proposed mechanistic link between BPD and PVS may be mediated, in part, by vascular endothelial growth factor (VEGF) and inflammation [3,6].

Together, severe BPD and PVS are a serious complication of prematurity with significant implications for prognosis [14]. Observational evidence is emerging that premature infants with PVS and either severe BPD and/or PH are at risk for decreased survival by close to 50% compared to premature infants alone [6,7]. Despite the increased recognition of PVS in premature infants with both pulmonary parenchymal and vascular disease, the unpredictable natural history of progressive PVS and difficulty in diagnosis make understanding the neonatal consequences challenging. Accordingly, in this review we explore the relationship between BPD and PVS, with a specific focus on the embryological impact of cardiopulmonary morphogenesis on venous development, and consequences of premature birth, epidemiology, clinical presentation, risk factors, and plausible biological mechanisms. We offer an algorithm for early detection and prevention and provide suggestions for research priorities.

## 2. Cardiac Morphogenesis and Airway Growth Influence Pulmonary Vascular Development

The pulmonary vasculature arises through temporal and spatially controlled signaling pathways linked to both the cardiac morphogenesis and airway development [15]. As the development of the lungs and pulmonary vasculature form concurrently during embryogenesis, there is an increased appreciation of the association of severe parenchymal lung diseases (e.g., BPD) and late and chronic PVD in premature individuals. As a vascular disorder, PVD is characterized by abnormalities in tone, histopathology, and unbalanced chemokines, but also reflects reciprocal abnormalities in parenchymal development. The PVS that presents in premature infants is not only a disease of venous obstruction to pulmonary blood flow but is also a manifestation of an underlying abnormality in total lung development, increasing the infant’s predisposition to intolerance of fluid in the alveolar space and acute respiratory failure. As such, it becomes even more imperative to understand the development of lung parenchyma and cardiac morphogenesis, as each relates specifically to pulmonary venous development and maldevelopment to comprehend the true mechanisms of PVS disease in premature infants (Figure 1).

### 2.1. Cardiac Morphogenesis

Cardiac morphogenesis precedes airway development during the embryonic stage. Critical components of the pulmonary system actually arise from the primitive heart tube [16]. The genetic investigation of cardiac morphogenesis has shown that the first heart field progenitor cells give rise to the left ventricle, and parts of the right and left atria. The second heart field progenitor cells mature into the right ventricle forming the main pulmonary artery trunk, parts of the atria, septum, and the base of the aorta during the looping process [15,17] (Figure 1). The cardiac mesoderm progresses from tube formation at two weeks, through asymmetric looping and septation into distinct chambers and outflow tracts by eight weeks. There are key genetic regulators that drive each of these processes throughout development [18].

### 2.2. Development of the Airways

Lung development begins around 4 weeks of gestation and proceeds through five stages: embryonic, pseudoglandular, canalicular, saccular, and alveolar (Figure 1) [19]. The embryonic period from weeks 4–8 of gestation is characterized by the growth of bronchial buds into the mainstem bronchi. The pseudoglandular period from weeks 8–17 of gestation comprises the branching of the airway buds. The canalicular stage from weeks 17–26 has the development of the lung capillaries and creation of the air–blood barrier. In the saccular stage from weeks 26–36, the last generation of airways develop terminating with clusters of thin-walled primitive alveoli, known as saccules. From 36 weeks through birth and beyond, the lungs are in the alveolar stage. Secondary septa grow outwards and subdivide the terminal saccules into anatomic alveoli. The microvasculature undergoes marked growth and development with much of the adult lung form and shape present by this stage.

### 2.3. Development of Pulmonary Vasculature

The branching airways and cardiac morphogenesis provide the scaffold for the developing vasculature. The size and number of arteries increase exponentially and consist of thin, elastic vessels that accompany the arborization of the bronchial airway [20]. The venous system also plays a special role in lung development. Early in the embryologic period, the peripheral mesenchyme contains only lakes or thin-walled vascular channels and an initial connection can be traced to a central patent vessel, the pulmonary vein. Their growth helps to establish an infrastructure within the developing lobes of the lung. Primitive venous structures are present and drain the networks of early vessels. The peripheral mesenchymal channels connect to the central vein while the pulmonary artery branches are still confined to the hilar region [21]. The pulmonary venous system is the first to establish a central connection with the heart [22]. The exact mechanisms of the pulmonary vein development are unclear, but a combination of secretory factors appear to drive the separation of the common pulmonary vein that invades the presumptive left atrium into four different orifices. A continuous circulation between the heart and lungs is present by the 5th week of gestation as the pulmonary vasculature precursors form a multilayered vascular network and mesenchymal capillary plexus linking the arterial and venous poles of the heart [23]. Throughout the pseudoglandular stage, preacinar pulmonary and bronchial arteries and veins develop around the growing airway buds and correspond to the bronchial branching pattern in the human lung by 17 weeks of gestation (Figure 1). Over the remainder of gestation, the vasculature continues to mature, thinning out and expanding, all in conjunction with airway, parenchymal, and cardiac growth. At birth, the overall pattern of the arteries and veins is the same pattern as in the adult. However, the vasculature continues to grow in complexity with time.

### 2.4. Vasculogenesis and Angiogenesis

The formation of the pulmonary vasculature is governed by two principal mechanisms: (1) vasculogenesis and (2) angiogenesis. Vasculogenesis is a de novo process: angioblasts or endothelial progenitor cells migrate and differentiate in response to local cues (e.g., growth factors and extracellular matrix) to vascular endothelial cells and form blood vessels. In contrast, in angiogenesis, new blood vessels arise by direct extension of pre-existing vessels. At different stages of gestation and in different locations within the lungs, each process likely drives development, rather than one process alone leading to the creation of vasculature. While pulmonary vascular development is regulated by the interplay between many different factors, vascular endothelial growth factor (VEGF) and transforming growth factor-β (TGF-β) are involved in the development of the pulmonary parenchyma. VEGF expression is regulated by hypoxia-inducible factors (HIF)-1 and 2, which are transcriptional complexes responding to changes in oxygen levels. Normal lung development takes place in the relatively hypoxic environment of the uterus which stabilizes the HIF complex, leading to the transcription of hypoxia-responsive target genes, such as VEGF. VEGF binds to two trans-membrane tyrosine-kinase receptors, VEGFR-1 (Flt-1) and VEGFR-2 (Flk-1), which are strongly expressed in endothelial cells. VEGF also plays an important role in epithelial branching morphogenesis and alveolar development. The impairment of pulmonary vasculature development by VEGF inhibitors in both fetal and newborn rats is accompanied by diminished alveolar development, which results in histologic findings resembling clinical BPD [24]. TGF-β plays a key role in epithelial and endothelial–mesenchymal interactions. Similar to VEGF interactions, TGF-β signaling is necessary for branching morphogenesis and pulmonary vascular development. In PVD, disturbed TGF-β signaling is the result of the BMPR2 (bone morphogenetic protein receptor II): a common genetic mutation associated with PH.

## 3. Consequences of Premature Birth on PVS

### 3.1. Parenchymal Lung Disease Influences Vascular Development

In premature infants, the hemodynamic signatures of PVS are influenced by pulmonary parenchymal and venous development. Since the lung capillary bed and air–blood barrier are not complete until late in the canalicular phase, and terminal air development and the alveoli are not formed until the saccular stage, infants born premature will have an arrest in vascular and alveolar growth in these early stages of airway development leading to heterogenous (decrease and enlarged) alveoli and a reduction in the number of capillaries as compared to a normal lung. This results in inadequate vascular growth, immature vascular function, and decreased host defense, which when combined with exposure to harmful stimuli after birth and through the neonatal period, propagate abnormal development of the lung circulation. For example, exposure to chronic mechanical ventilation following premature birth is associated with vascular changes characterized by intimal proliferation, simplification of alveolar structures, and pathologic vessel growth in response to inflammation. The effect of hypoxia and hyperoxia on immature circulations also encourage an increase in arterial and venous smooth muscle. Venous resistance is greatest at birth and drops quickly with postnatal changes in oxygen tension. Premature infants with BPD that develop chronic PH show a greater increase in pulmonary vascular smooth muscle than those without PH [24]. Echocardiography studies completed at birth, well before chronic pulmonary parenchymal or vascular diseases of prematurity (BPD or chronic PH, respectively) can be diagnosed, do not typically show PVS. However, with shared risk factors for BPD and PVS, the developmental underpinnings and anatomical considerations for each disease may already be in motion.

Oxygen tension is an important component of lung and vascular development with aberrations contributing to the pathogenesis of both BPD and PVD. Unfortunately, through the perinatal period, neither hypoxia nor hyperoxia produce favorable conditions for continued vascular development [23]. The neonatal pulmonary circulation is extremely reactive, proliferative, and matrix-producing. When exposed to hypoxic, hyperoxic, or toxic stimuli, neonatal animals exhibit dramatic structural changes through cell proliferation and matrix-protein accumulation, resulting in rapidly progressive and potentially less reversible PH and right ventricular hypertrophy [25]. The effects of abnormal oxygen tension may produce effects on the pulmonary venous circulation that can last beyond infancy.

### 3.2. Immature Cardiac Development

The intrinsic features of the developing myocardium also place the premature infant with PVS at risk of further hemodynamic compromise [26]. The immature myocardium consists of an underdeveloped contractile organization with fewer mitochondria and myofibrils, inadequate compliant collagen, and a calcium handling system that is not fully formed. Premature birth interrupts cardiomyocyte proliferation and reduces cardiomyocyte endowment compared to term-born infants, further complicating physiological structural remodeling in the neonatal period. Although reduced heart mass following premature birth can be normalized by disproportionate cardiac hypertrophy and increased left ventricular mass, the loss of cardiomyocyte endowment is a key factor influencing susceptibility to cardiopulmonary disease [27]. As such, the preterm myocardium is poorly tolerant to increases in afterload according to the force–frequency relationship, lacks the reserve to cope with states of reduced preload, and is prone to diastolic dysfunction with changes in the Frank–Starling curve [28]. In premature infants, the interruption of placental blood flow in utero and subsequent increased pulmonary blood return on an underdeveloped left heart may leave it incapable of acutely adapting to volume loading conditions over time and predispose these group of infants to PVS as well, especially in the setting of a left-to-right cardiac shunt [1,7]. This phenomenon is even more pronounced in the premature infants with fetal growth restriction (FGR). In the later neonatal period, the impairment of diastolic inflow of the pulmonary vein into the left atrium will alter the Frank–Starling curve and leave the further negative impact of cardiopulmonary interactions [29]. Additional insults, such as the persistence of fetal shunts (ductus venosum, ductus arteriosus, or foramen ovale); adverse effects of ventilation strategies; and pharmaceutical intervention (e.g., perinatal steroids) can further compromise the immature myocardium during the critical window of development over the first year of age, alter normal maturation, and contribute to the unique cardiac phenotype of prematurity [30].

### 3.3. Epidemiology of PVS in Premature Infants

The prevalence of PVS in premature infants has increased over the past decade as more individuals are surviving beyond the neonatal period, coupled with improved methods of detection of disease. Several studies have examined the prevalence of PVS in premature infants with BPD [14,31,32,33]. The median gestational age of infants in the studies ranged from 25–27 weeks at birth. In infants with BPD alone, the prevalence of PVS was approximately 4.7%, and increased to 23–27% when infants who had BPD and chronic PH were included. However, the degree to which prematurity contributes to the development of PVS without BPD as a comorbidity is still less clear. In one study of infants with PVS, 50% of infants were born between 32 and 37 weeks, and 37% of infants were born at <28 weeks, suggesting that, although the most premature infants are at the highest risk for BPD, the association between BPD and PVS may be complex and dependent on additional factors [9,34]. In another study of premature infants with BPD undergoing cardiac catheterization for evaluation of PH, PVS was detected in only 23% of patients [31]. The strength of the association of BPD with PVS seems to differ based on whether the populations being studied are infants with BPD alone or infants with PVS, suggesting not all infants with severe BPD develop PVS, but infants with PVS often have a clinical history of BPD. In studies of infants who have a diagnosis of PVS, between 34 and 88% of infants have BPD, most commonly severe BPD, as a comorbidity [7,9,11,32,34,35,36]. Interestingly, 25–40% of infants with severe BPD develop PH, which is thought to be secondary to pulmonary vascular remodeling and neointimal proliferation following injury to the developing lung. More than half of infants with PVS also have a diagnosis of PH, raising the possibility that PVS is an important cause of PH in infants, or that there may be an ascertainment bias in PVS detection [2,9]. Taken altogether, an accurate depiction of the epidemiologic profile of PVS in premature infants is still precluded by inconsistent surveillance protocols and disease definitions [6].

Interestingly, between 34 and 61% of all infants with PVS are premature [9,35,36,37], with the median age at diagnosis in premature infants ranging from 5 to 7.5 months of age, and most studies reporting diagnosis at greater than 6 months of age (Figure 2) [2,32,34,35,37]. In comparison, the diagnosis is slightly earlier for infants with cardiac defects, ranging between 4 and 7 months [9]. Neonates with PVS diagnosed before 6 months of age, regardless of the presence of prematurity or CHD, have nearly a three-fold increase in mortality [9], with a significant survival benefit with older age at diagnosis [7,38]. PVS left untreated can be a lethal anomaly with poor long-term prognosis and high mortality [12,35].

## 4. BPD Is an Inflammatory State: Relevance to PVS

The pathogenesis of PVS suggests that the same processes that play a role in the development of BPD, some of which are inflammatory mediated, may also be responsible for the postnatal development of PVS. BPD is a consequence of disrupted lung development in both the airway and the vasculature and has been described as an inflammatory state of prematurity. Phenotyping preterm inflammation and BPD can be divided into three phases from pregnancy through perinatal transition, and resuscitation, and the neonatal course (Figure 3). During pregnancy, the placenta faces several pro-inflammatory perturbations including infection, chorioamnionitis, genetics, and even epigenetic processes. At birth, premature lungs may need mechanical ventilation, which subsequently places them at risk for oxidative stress and toxicity, and even barotrauma. Throughout the NICU course, ongoing exposures to mechanical ventilation can be further complicated by difficulty in meeting nutritional goals as well as exposure to common comorbidities of prematurity. Collectively, these factors promote the release of a multitude of inflammatory markers and further contribute to the development of BPD. The “classic” or “old” BPD is heavily influenced by lung injury, inflammation, and fibrosis; meanwhile, “new” BPD is a function of arrest of alveolar development, or vascular remodeling.

The risk for the development of BPD increases with decreasing gestational age, and many neonatal and postnatal factors that contribute to the development of severe BPD have also been linked to the development of PH and PVS in infants with BPD [39,40,41]. However, understanding the inflammatory interactions between BPD and PVS begins with elucidating the cellular and molecular mechanisms implicated across the spectrum of PVS [42]. At the cellular level, the key pathologic change driving intraluminal stenosis is intimal proliferation in the vessel wall. Fibroblast and myofibroblast proliferation and matrix deposition within the pulmonary vein walls lead to a markedly thickened intima. At the molecular level, the vascular endothelial growth factor or VEGF signaling pathways have been identified as targets of interest in pulmonary venous pathology [43].

As mentioned, there is an established relationship between BPD and PH in extreme premature infants, and clinically, the infants with both severe BPD and PH can develop PVS [7]. While inflammation can play a role in each disease entity, the link between BPD, PH, and PVS is less well-described. Extensive pathological analyses of lung samples from children with PVS have demonstrated that these infants exhibit muscularization of large intrapulmonary veins and pulmonary artery remodeling characteristics of those seen with chronic PH. Recently, several investigators have postulated that there is a mechanical stretching/traction effect that occurs with the BPD-PH phenomenon [1,44]. We know there is hyperinflation and fibrosis in patients with BPD, and this may lead to mechanical stretching/traction of the pulmonary veins. Zhang et al. recently showed with an animal model that the extracellular matrix of the vein experiences stretch via the TGF-β1 receptor, and intracellular signaling in vessel fibroblasts causes differentiation to the myofibroblast [44]. This process continues to cause further stretch/traction on extracellular matrix, ultimately leading to significant myofibroblast proliferation and PV obstruction.

The pulmonary veins pass from the lung parenchyma to the left atrium, frequently taking a long, angulated course around abutting extra-pericardial structures [1] (Figure 1). As such, some investigators have also postulated that the pathogenesis of PVS may be related to turbulent blood flow and vascular remodeling [45]. The vessel course can exacerbate wall shear stress, increase blood flow turbulence, and promote vascular fibroblastic remodeling, which may contribute to the development of PVS. We suspect this may be further exacerbated in patients with BPD. Recently, investigators used novel approaches to assess wall shear stress in children with PVS and showed that interventions on the veins can help reduce the wall shear stress and potentially remove the trigger for stenosis progression [45].

Premature infants are also at risk for chronic aspiration, and beyond exacerbating the different phenotypes of BPD, it has recently been shown that PVS infants with clinical aspiration had nearly five times higher odds of poor treatment response than patients without aspiration [46]. Aspiration may decrease lung elastance and lead to airway hyperinflammation, which can cause traction on the pulmonary veins and increase the shear stress on the vessel walls. This process will promote the differentiation of fibroblasts to a myofibroblast and lead to intraluminal cell proliferation and recurrent PVS [46].

## 5. Additional Neonatal Risk Factors

The neonatal risk factors for PVS, including prematurity and its related-comorbidities [e.g., necrotizing enterocolitis (NEC) [7,12], persistent left-to-right shunt physiology (e.g., patent ductus arteriosus, PDA and ventricular septal defects, VSD) [35], FGR) [1,12,35,47], cardiac defects, and genetic and epigenetic predisposition [7] may each contribute independently to the development and progression of disease. However, considerable overlap may also be present between risk factors, as infants with cardiac defects are more likely to be born prematurely, or infants with genetic predispositions are more likely to have cardiac defects (Figure 2). Although the intersection between predisposing conditions is not clearly defined, infants with multiple risk factors should be considered at higher risk for the development of PVS and must be screened commensurate with clinical suspicions.

### 5.1. Necrotizing Enterocolitis

There is an established relationship between prematurity, BPD, and NEC [48]. Studies of premature infants with BPD have also shown that 23–63% of infants with PVS have a co-diagnosis of NEC, which is substantially higher than the 9% incidence reported in the premature population [7,9,12,32,49]. This increased prevalence of NEC in the PVS population suggests that inflammatory cascades that occur with intestinal injury may play a role in the pathogenesis of PVS. There also may be a link between NEC and PVS via similar VEGF-mediated inflammation processes that occur with BPD [12,50,51]. However, in one study, the NEC rate in infants with PVS was the same as it was for infants with BPD and PH, complicating the possibility of a causal relationship between NEC and PVS [32].

### 5.2. Left-to-Right Shunt Physiology

Over half of the infants with PVS have left-to-right shunt lesions, such as a VSD or a PDA, which supports the hypothesis that embryologic alterations and increased pulmonary blood flow may contribute to its development [1,2,11]. Infants with PVS have associated isolated ASDs 13–64% of the time, isolated VSDs 15–31% of the time, and complete AV canal defects 9–13% of the time [7,9,11,35,36]. It has been proposed that a persistent left-to-right PDA during the neonatal course may also predispose infants to PVS, with the proportion of infants with PDA and PVS ranging from approximately 4 to 59%, with a median prevalence of 50% [7,9,11,35,36]. Interestingly, infants with BPD have a PDA 20–37% of the time, but early medical or surgical closure have not always been associated with benefits to neonatal outcomes or a decrease in PVS [14,31,32,52].

The duration of left-to-right shunts may also affect the likelihood of developing PVS. In a study of 23 infants with Trisomy 21, the median duration of left-to-right shunt in infants with PVS was 6 months, compared with 3 months for those infants without PVS [53]. Additionally, 73% of infants with PVS had their primary repair at >4 months, compared with 47% of infants without PVS. Of the infants in this cohort, most had associated congenital heart disease (complete AV canal), 74% had multi-vessel pulmonary vein involvement, and 60% had PH. In multivariable analyses that adjusted for shunt duration and gestational age, each month of exposure to a left-to-right shunt in this population increased the odds of PVS by 1.2 (95% CI 1.06–1.39). Interestingly, among infants who were born at <35 weeks’ gestation at birth, the odds for developing PVS were even higher at 4.8 (1.4–16.8), potentially suggesting an exponential escalation of risk depending on the presence of prematurity or the number of risk factors present [53].

### 5.3. Fetal Growth Restriction

Infants with a birthweight less than the third percentile for gestational age, who have FGR at birth, have been shown to be at a higher risk of developing severe BPD and PH [54]. FGR has also been associated with PVS, suggesting that fetal programming and the intrauterine environment may play a role in its development. In a study of 213 infants with severe BPD, 11% of infants were growth-restricted at birth, whereas 80% of infants with BPD and PVS were growth-restricted [14]. Having a diagnosis of PVS, and being growth-restricted, when compared with being appropriately grown for the gestational age at birth, seem to confer an increased risk for mortality as well. At 2 years of age, approximately 40% of infants who had a growth restriction and PVS survived, compared with a >70% survival probability for appropriately grown infants [7].

### 5.4. Genetic Linkage

Aneuploidy or genetic syndromes are associated with an increased risk for PVS [36]. In a systematic review of PVS literature, 9% of infants had associated genetic conditions, with most infants having Trisomy 21 [2]. In a large PVS registry, 157 infants had genetic data evaluated to aid with the clinical phenotyping of patients [36]. Of those with genetic testing information available, 32% had known genetic variants and 33% had variants of unknown significance. Of the 23 infants with known genetic variants, the most common genetic abnormalities were Trisomy 21 (57%), Smith–Lemli–Opitz syndrome (22%), and DiGeorge syndrome (9%), followed by Cat-Eye syndrome, Mosaic Turner’s Syndrome, and Adams–Oliver syndrome-5 (<5% each).

PVS in infants with Trisomy 21 has been found to be rapidly progressive, often associated with PH, and carries a high mortality rate within a few years of diagnosis [55]. The progression of PVS in Trisomy 21 patients appears to be more aggressive than other neonates [1]. Similarly, Trisomy 21 infants are also at an increased risk of having more rapid progression of PH compared to the general population [56,57], in part due to abnormalities in pulmonary vasculature development, and impairments in the regulation of vascular tone and vascular endothelial function [58]. On the other hand, infants with Smith–Lemli–Opitz are often diagnosed earlier, between birth and 2 months of life, and have a clinical course that rapidly progresses from single vein to multi-vein involvement. Infants with phenotypes suggestive of Smith–Lemli–Opitz also represent a high-risk population for the development of PVS and should undergo further genetic and cardiac evaluations.

A heritable component to the pathogenesis of PVS is suggested by a genome-wide linkage analysis of four affected siblings who presented with primary prenatal lymphatic abnormalities and who were found to have a locus for the defect mapped to chromosome 2q35-2q36.1 [59]. An additional case series of three affected siblings with unilateral pulmonary vein atresia supported a potential genetic contribution to the development of PVS [60].

### 5.5. Twin Pregnancy

Infants with PVS, who are products of multiple-gestation pregnancies, have an unaffected twin 39% of the time, suggesting that factors other than genetic predisposition may also play a role in the development or progression of PVS [7].

### 5.6. Retinopathy of Prematurity

PVS has been reported to occur with higher incidence in neonates with retinopathy of prematurity (ROP) [7,9,35]. As another process due to disorganized vascular development mediated by VEGF, ROP presents in the later neonatal period. Premature birth and hypoxia alter VEGF signaling and arrest the vasculature growth in the eye in the early neonatal period. In the later neonatal period, with less metabolic demand and less oxygen requirements, VEGF signaling increases. This time frame mirrors the development of PVS.

## 6. Clinical Presentation of PVS in Premature Infants

The pathophysiologic consequences of PVS are largely determined by the number and severity of stenosed vessels [61]. As such, there is a wide range of clinical presentations of premature infants with PVS, including persistent and frequent hypoxemia and respiratory distress, prolonged supplemental oxygen need, inability to wean from respiratory support, and/or new onset PH associated with chronic lung disease [62]. The initial signs and symptoms are often non-specific and are also encountered in preterm infants with PH and BPD alone. In addition to tachypnea, increased work of breathing, and retractions, preterm infants with PVS may also experience new or worsened PH, failure to maintain or gain weight, or require unexplained increases in ventilatory or oxygen support beyond the expected clinical trajectory [3].

In premature infants, the narrowing of the pulmonary veins presents at different stages during the neonatal course characterized by the degree of pulmonary venous congestion, interstitial edema, and upstream elevation of PAP. Mean PAP (mPAP) is quantified by the relationship of pulmonary vascular resistance (PVR), pulmonary blood flow (PBF), and pulmonary capillary wedge pressure (PCWP) and derived from the equation: mPAP = [(PVR × PBF) + PCWP] (Figure 4). Perturbations to any of these contributors can add an extra stressor to the immature pulmonary vasculature, predisposing it to vascular remodeling and the potential development of chronic PH. Although the rise in mPAP is most associated with severe BPD and chronic vascular remodeling in premature infants secondary to changes on PVR or increased PBF, PVS is also recognized as a cause of pulmonary venous congestion [37]. As PVS obstructs blood flow back to the left atrium, it results in elevated pulmonary venous pressure. The PH that occurs in the neonatal period with individuals with PVS can also be due to a single stenotic pulmonary vein complicated by BPD, suggesting a multifactorial and heterogeneous pathophysiology [34,35]. PVS must be considered in those premature infants who demonstrate worsening PH despite optimization of cardiopulmonary support, recurrent pulmonary edema, or even poor responsive or adverse effects of PH-targeted therapy with pulmonary vasodilation leading to capillary flooding and interstitial edema [10].

The progression of clinical disease is also impacted by additional vein involvement, despite treatment. The number and location of impacted pulmonary veins (≥3 and bilateral), distal/ upstream disease (degree of elevated pulmonary pressures and right ventricular dysfunction), and restenosis after intervention are also linked to mortality, that is further complicated in preterm-born individuals. Bilateral, multiple pulmonary vein involvement, and recurrent stenosis are the clinical expressions of an aggressive disease process and, therefore, a worse survival [7,63]. The extension of acquired intraluminal PVS into multiple vessels can also occur with any of these etiologies, but the presence of multi-vessel involvement is a far more worrisome sign with an extremely poor prognosis [64,65]. Infants with a greater number of veins affected are at the highest risk for morbidity and mortality [9]. Holt et al. demonstrated from the Pediatric Cardiac Care Consortium that bilateral vessel involvement was predictive of lung death, defined as death or lung transplantation [62].

While BPD and chronic PH can partially improve or even be attenuated with enhanced nutrition, optimization of ventilator strategies, and overall parenchymal and vascular growth during the neonatal and postnatal period, postcapillary PH due to PVS has a substantial risk of recurrence or progression that can clinically present as right ventricular failure leading to high mortality [11,61]. Indeed, these aspects specific to PVS—recurrence locally and progression locally and regionally—make this disease uniquely challenging. The recurrence and progression do not respond to typical vascular interventions (such as surgery or catheter-based therapies), but they appear to require systemic therapies to alter or reset disrupted cellular pathway signals [66,67].

## 7. Challenge of PVS Detection in the NICU: Algorithm for Detection PVS in Premature Infants

The heterogenous presentation of PVS in premature infants and lack of universal screening protocols historically posed significant challenges in early diagnosis [6]. However, improved understanding of the pathophysiology of PVS and its associated risk factors of prematurity (e.g., BPD, NEC, ROP, PVD, T21) have forced health care providers to remain vigilant to the potential contributions of PVS with a low threshold for screening. In view of its widespread availability, non-invasive nature, and non-reliance on sedation, transthoracic echocardiography (TTE) remains the first-line imaging technique in the evaluation of pulmonary venous disease [2]. TTE allows the assessment of premature infants on high frequency ventilatory support or in patients that cannot be transported to obtain other imaging studies. This bedside procedure can be remotely interpreted by clinicians, allowing for its use in nurseries in rural areas. Echocardiography also allows for the assessment of hemodynamics.

Screening for PVS in high-risk populations of premature infants has recently been incorporated into expert guidelines and society-endorsed protocols for associated severe PH and BPD [10,68,69]. The Pediatric Pulmonary Hypertension Network recommends screening guidelines for PH in premature infants requiring ventilator support at or ≥7 days or 36 weeks postmenstrual age with moderate to severe BPD [68]. Similarly, guidelines from the American Heart Association and American Thoracic Society for evaluation of pediatric pulmonary hypertension also advocated for earlier screening of premature infants with severe respiratory distress syndrome, a high supplemental oxygen requirement, or ventilatory support and prematurity <26 weeks [69]. Collectively, these guidelines specifically recommend detailed echocardiogaphic evaluations for peripheral pulmonary artery stenosis, pulmonary vascular disease, and pulmonary vein stenosis. This has led to the development of detailed screening protocols for PH in various neonatal units nationwide, and has allowed for the detection of acquired pulmonary vein stenosis [70].

Although TTE is easily accessible in most patients and allows bedside assessment of critically ill infants, it has several limitations. Acoustic windows may be inadequate in some premature infants, such as those with chronic lung disease, limiting the diagnostic yield of TTE. Not surprisingly, a recent population study observed that 26% of PVS cases were missed on TTE and subsequently diagnosed at cardiac catheterization [37]. Furthermore, a separate report showed that of 26 premature infants studied by cardiac catheterization to characterize their chronic PH, PVS was diagnosed in 7 (27%), but had been suspected by echocardiography in only 3 of the 7 cases [71]. Moreover, patient-intolerance (oxygen desaturations, temperature instability) has led to growing interest in the use of limited TTE studies, wherein standard imaging planes and the 2-dimensional imaging and color Doppler of all veins are not performed [2]. However, as described by Minich et al. the risk of missing a diagnosis of primary PVS was eight-fold higher among infants who received a limited TTE (2D imaging and color Doppler of all veins) versus a complete TTE [72]. Finally, an echocardiography is sometimes prohibitive as a screening tool with high false negative and false positive rates [70]. False positive rates with high pulmonary vein velocities and no anatomic PVS can be often seen in lesions with significant left-to-right shunts. With PVS being a progressive disease, it may not be identified on repeat echocardiograms for other indications due to a lack of pulmonary vein evaluation contributing to a high false negative rate. These observations led the Pediatric Pulmonary Hypertension Network to emphasize that, in the context of concerns for PH, pulmonary vein flow and gradients should be examined as part of each study [68].

Echocardiographic-based scoring systems that evaluate PVS disease severity have emerged to guide interventions and predict outcomes [65]. Despite growing interest in their development and validation, limitations persist in the use of TTE-based scoring systems in the accurate diagnosis of PVS [10]. While echocardiography can demarcate “downstream” PVS at the veno-atrial junction, “upstream” stenosis is obscured by lung parenchyma [73]. Moreover, in the setting of progressive stenosis, flow redistribution from parenchymal lung segments with obstructed veins to unobstructed segments may decrease echocardiographic-derived gradients. Thus, leading investigators and governing bodies have suggested that comprehensive evaluations of the pulmonary veins in premature infants with PH should include cardiac catheterization, computed topography angiography, or cardiac magnetic resonance imaging for a more detailed characterization of vessel involvement [10]. Finally, with recognition that less than 20% of PVS is typically detected before discharge from the neonatal intensive care unit, and most infants have a median of 3–5 echocardiograms before diagnosis, an understanding of neonatal risk factors that contribute to its development are important to determine appropriate post-discharge surveillance strategies.

Due to the pitfalls of echocardiography as mentioned above, increasing the use of CT scans as a screening tool has been advocated [74]. The current estimates of ionizing radiation exposure from cardiac CT scans for children <1 year are 0.2–9.6 mSV^7^. Due to the concern for excessive ionizing radiation exposure of infants in the NICU, Scott et al. assessed 215 premature infants and found that 12.1% of neonates <33 weeks’ gestation exceeded the recommended radiation exposure in their cohort primarily related to central line placements and gastrointestinal evaluations [75]. The judicious use of cardiac CT scans should be considered to avoid the harmful effects of radiation exposure. CT scans and other imaging studies’ availabilities are often limited to tertiary care centers due to the need for skilled personnel to perform and interpret these studies. In certain patients, the need for sedation limits obtaining a CT scan.

Although diagnostic evidence-based guidelines do yet exist for preterm infants with BPD, most experts agree there is a need for developing robust protocols specifically aimed at the recognition of risk factors, awareness of clinical signs and symptoms, and utilization of echocardiography as a screening tool for identification of PVS. In high-risk patients or those identified by echocardiography, further evaluation with CT scans, lung perfusion scans, or cardiac catheterization should be considered/performed as is currently done at some centers across the country [70]. Finally, consultation with care providers and institutions that have expertise in caring for children with PVS is highly recommended to develop an individualized treatment and follow-up plan.

## 8. Conclusions and Future Research

Primary PVS is a rare disease with high morbidity and mortality in preterm-born individuals. Despite limited numbers of premature infants available for study in each individual center, as well as striking inter and intra-site variability in disease management, multi-center collaborations (e.g., PVS network) have begun to shed light on understanding the natural history of PVS in this high-risk population. Premature-born individuals are now recognized as a special population of individuals at risk to develop PVS with a unique disease trajectory dictated by embryological considerations, comorbidities, hemodynamic signatures, and vein involvement (Figure 5) [42]. In this review, we highlighted the evidence on the pathobiology, pathophysiology, and optimal diagnostics strategies of premature infants with PVS. While the role of conservative, therapeutic, catheterization-based, and surgical intervention for primary PVS in premature infants was beyond the scope of this review and has been presented elsewhere [3,6], increased recognition of the epidemiology, developmental underpinnings, anatomical considerations, risk factors, clinical consequences, and diagnostics considerations of PVS in premature infants will serve to guide the continued preclinical and clinical investigations into the development of evidence-based management approaches.

The high mortality rate that is seen with severe BPD, PVD, and PVS requires further minimization of parenchymal and vascular disease beyond the perinatal period. This may be achieved by modifying postnatal comorbidities such as chronic respiratory failure, poor growth, and infection. In addition, utilizing registry data at the multi-center level will only go so far; multidisciplinary teams at the center-level must include neonatology, cardiology, pulmonology, interventionalists, and surgeons to identify and address risks for further parenchymal and pulmonary vascular disease. Furthermore, large comparative effectiveness studies to determine best practices must also be developed with a focus on premature infants through rigorous phenotyping and endotyping. With increased recognition of the lack of specific diagnostic protocols and center variation, future efforts must clearly define the diagnostic criteria needed to improve early detection and guide management decisions in the context of various gestation ages. As contemporary definitions fail to reflect that PVS in premature infants is likely a diverse disease marked by multiple, poorly characterized phenotypes, including a variable amalgamation of the numbers of involved veins, levels of stenosis (discrete, diffuse, multifocal), and consequent clinical manifestations, we need more nuanced approaches based on objective disease phenotyping to offer significant promise for improving evidence-based care.

## Figures and Tables

**Figure 1 children-09-00713-f001:**
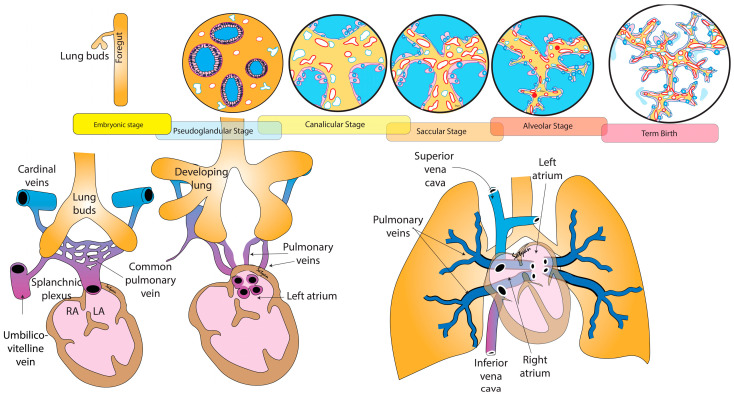
Development of pulmonary veins in relation to lung and cardiac development. The emerging lung buds are surrounded by splanchnic plexus connected to the umbilico-vitelline vein and the right and left cardinal veins. The pulmonary venous system drains to the left atrium through one common vein. Subsequently, there is organization of the venous network leading to four individual pulmonary veins draining into the left atrium. Copyright Satyan Lakshminrusimha.

**Figure 2 children-09-00713-f002:**
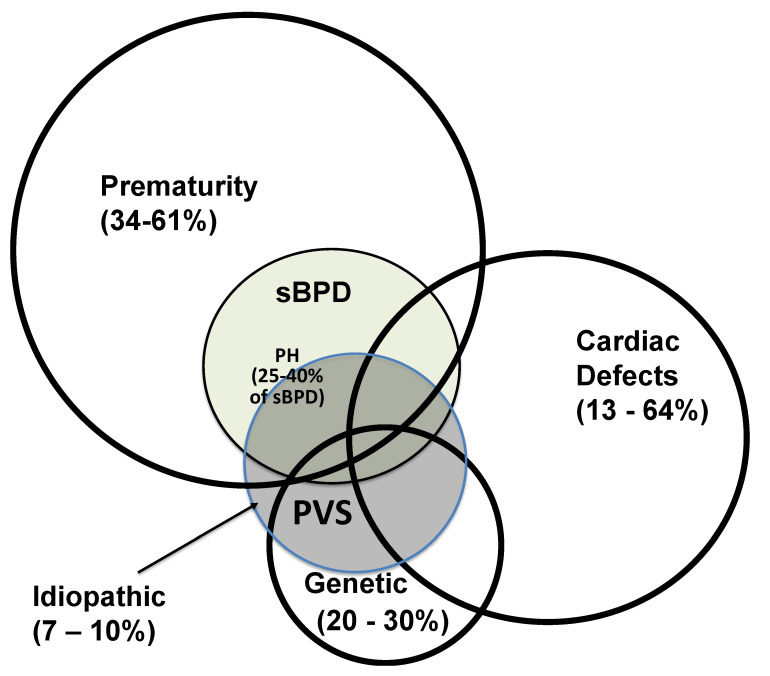
Epidemiological associations with pulmonary vein stenosis (PVS—blue central circle). sBPD—severe bronchopulmonary dysplasia (green). PH—pulmonary hypertension. Infants with sBPD and PVS are at risk for PH (light-blue). See text for details.

**Figure 3 children-09-00713-f003:**
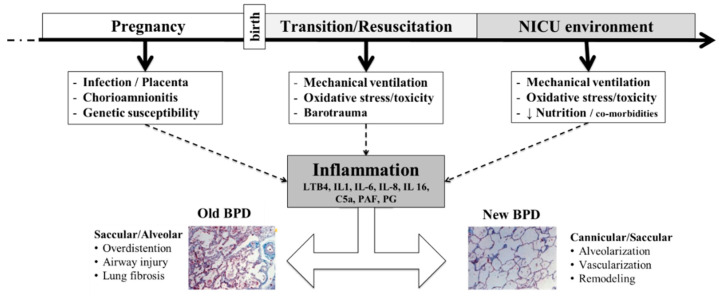
The three phases of lung inflammation contributing to bronchopulmonary dysplasia in preterm infants.

**Figure 4 children-09-00713-f004:**
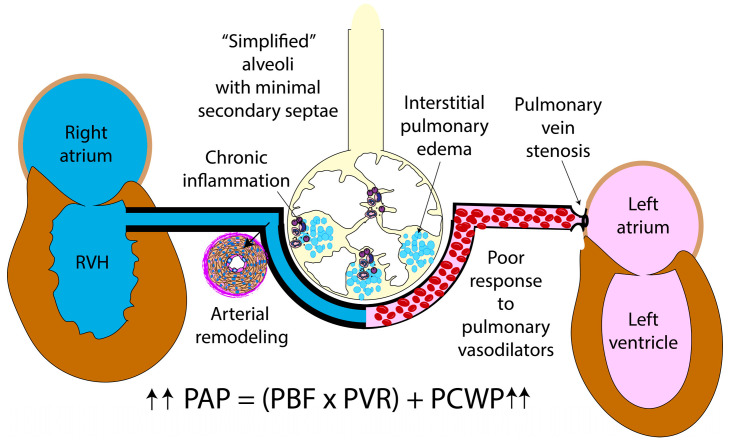
Hemodynamics in pulmonary vein stenosis. Pulmonary venous congestion results in interstitial edema leading to poor response to pulmonary vasodilators. Elevation of pulmonary capillary wedge pressure (PCWP) leads to elevated pulmonary arterial pressure (PAP). Further alterations in pulmonary blood flow (PBF), as seen in left-to-right shunts, or changes in pulmonary vascular resistance with BPD and chronic PH can further elevate PAP. Copyright Satyan Lakshminrusimha.

**Figure 5 children-09-00713-f005:**
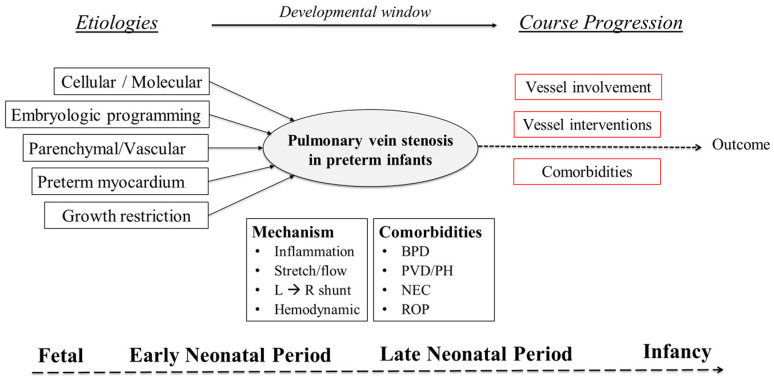
Preterm infants are a unique population of PVS: course progression.

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
