# Peer review of "Prematurity and Pulmonary Vein Stenosis: The Role of Parenchymal Lung Disease and Pulmonary Vascular Disease"

_children, 2022, doi:10.3390/children9050713_

Round 1

Reviewer 1 Report

Thank you for submitting your work.

I have some comments and questions that the authors may consider.

#1. The authors should start by stating the definitions of PVS and BPD in this review

#2. The authors should have a new part on the specific treatment for PVS.

#3. In the clinical presentation part, the author should describe the symptoms of PVS infants in more detail.

#4. The authors should describe more specific protocols for transthoracic echocardiography, CT angiography, and cardiac catheterization to improve the detection of PVS.

Author Response

Reviewer 1:

Thank you for the helpful suggestions. We have provided answers below according to the four commnents.

Comment 1. The authors should start by stating the definitions of PVS and BPD in this review.

Response 1: Thank you for this valuable comment. We have expanded the first paragraph to include definitions of PVS and BPD.

 PVS: Pulmonary vein stenosis (PVS) is a heterogeneous disease process that contributes to morbidity and mortality in infants and young children [1,2]. PVS describes the pathologic process of intraluminal narrowing of the veins that carry oxygenated blood from the lungs back to the left side of the heart [3].

BPD: As a morphologic disruption of all components of the lung, including airway, vascular, and lymphatics features, BPD is characterized by development of simplified alveolar structures, pathological vessel growth and remodeling in the pulmonary arterial and venous beds. While several approaches exist to define BPD in preterm infants, all focus on the need for prolonged oxygen therapy and/or respiratory support [8].

Comment 2. The authors should have a new part on the specific treatment for PVS. 

Response: Thank you for this comment. The treatment for PVS is an evolving field of science and has been addresses in other excellent manuscripts in this series. We have, therefore, referenced these other open access manuscripts that extensively describe different treatment approaches for PVS. Please see page 14, section 8.

Comment 3. In the clinical presentation part, the author should describe the symptoms of PVS infants in more detail.

Response: Thank you for these important comments. We have expanded the clinical presentation section to include additional the symptoms of PVS. Please see page 11, section 6.

Clinical Presentation of PVS in Premature Infants

The pathophysiologic consequences of PVS are largely determined by the number and severity of stenosed vessels [69]. As such, there is wide range of clinical presentations of premature infants with PVS, including persistent and frequent hypoxemia and respiratory distress, prolonged supplemental oxygen need, inability to wean from respiratory support, and/or new onset PH associated with chronic lung disease [70]. The initial signs and symptoms are often non-specific and are also encountered in preterm infants with PH and BPD alone. In addition to tachypnea, increase work of breathing, and retractions, preterm infants with PVS may also experience new or worsened PH, failure to maintain or gain weight, or require unexplained increases in ventilatory or oxygen support beyond the expected clinical trajectory [3].

Comment 4. The authors should describe more specific protocols for transthoracic echocardiography, CT angiography, and cardiac catheterization to improve the detection of PVS. 

Response: Thank you for this comment. Currently, there is no unifying evidence-based national or international protocol for detection with variation existing amongst centers. Clearly defined diagnostic criteria and more data in various gestation age is needed (as highlighted by reviewer 2).

We have added language in the section: Challenge of PVS Detection in the NICU: Algorithm for detection PVS in Premature Infants.

Although diagnostic evidence-based guidelines do yet exist for preterm infants with BPD, most experts agree there is a need for developing robust protocols specifically aimed at recognition of risk factors, awareness of clinical signs and symptoms, and utilization of  echocardiography as a screening tool for identification of PVS. In high-risk patients or those identified by echocardiography, further evaluation with CT scans, lung perfusion scans, or cardiac catheterization should be considered/performed as currently done at some centers across the country [81]. Finally, consultation with care providers and institutions that have expertise in caring for children with PVS is highly recommended to develop an individualized treatment and follow-up plan.

We have added additional language in the Conclusion and Future Research section to reflect this important point.

With increased recognition of the lack of specific diagnostic protocols and center variation, future efforts must clearly define the diagnostic criteria needed to improve early detection and guide management decisions in the context of various gestation ages.

Reviewer 2

We thank Reveiwer two for helpful comments

The article entitled “Prematurity and pulmonary vein stenosis: The role of parenchymal lung disease and pulmonary vascular disease” has been reviewed. The article was well written and showed detail description and discussion of the development of PVS and the relation of BPD and PVS. It is a new concept and require further investigation. 

Comment 1: I would suggest that the authors provide more detail definition of pulmonary vein stenosis, and the correlation between PVS and gestation age of prematurities.

Response 1:  Thank you for these comments. We have provided a more detailed definition of PVS. We agree that defined diagnostic criteria and more data in various gestation age would be very helpful in the future research – and as such have added this to our conclusion and future research section.

Introduction. First paragraph: Pulmonary vein stenosis (PVS) is a heterogeneous disease process that contributes to morbidity and mortality in infants and young children [1,2]. PVS describes the pathologic process of intraluminal narrowing of the veins that carry oxygenated blood from the lungs back to the left side of the heart [3].

Comment 2: Clearly defined diagnostic criteria and more data in various gestation age would be very helpful in the future research.

Response 2: We have provided language below to reflect this comments. Please comment 4 from Reviewer 1

Conclusion and Future Research. Last Paragraph: With increased recognition of the lack of specific diagnostic protocols and center-variation, future efforts must clearly define the diagnostic criteria needed to improve early detection and guide management decisions in the context of various gestation ages.

Reviewer 2 Report

The article entitled “Prematurity and pulmonary vein stenosis: The role of parenchymal lung disease and pulmonary vascular disease” has been reviewed. The article was well written and showed detail description and discussion of the development of PVS and the relation of BPD and PVS. It is a new concept and require further investigation. I would suggest that the authors provide more detail definition of pulmonary vein stenosis, and the correlation between PVS and gestation age of prematurities. Clearly defined diagnostic criteria and more data in various gestation age would be very helpful in the future research.

Author Response

Reviewer 1:

Thank you for the helpful suggestions. We have provided answers below according to the four commnents.

Comment 1. The authors should start by stating the definitions of PVS and BPD in this review.

Response 1: Thank you for this valuable comment. We have expanded the first paragraph to include definitions of PVS and BPD.

PVSPulmonary vein stenosis (PVS) is a heterogeneous disease process that contributes to morbidity and mortality in infants and young children [1,2]. PVS describes the pathologic process of intraluminal narrowing of the veins that carry oxygenated blood from the lungs back to the left side of the heart [3].

BPD: As a morphologic disruption of all components of the lung, including airway, vascular, and lymphatics features, BPD is characterized by development of simplified alveolar structures, pathological vessel growth and remodeling in the pulmonary arterial and venous beds. While several approaches exist to define BPD in preterm infants, all focus on the need for prolonged oxygen therapy and/or respiratory support [8].

Comment 2. The authors should have a new part on the specific treatment for PVS.

Response: Thank you for this comment. The treatment for PVS is an evolving field of science and has been addresses in other excellent manuscripts in this series. We have, therefore, referenced these other open access manuscripts that extensively describe different treatment approaches for PVS. Please see page 14, section 8.

Comment 3. In the clinical presentation part, the author should describe the symptoms of PVS infants in more detail.

Response: Thank you for these important comments. We have expanded the clinical presentation section to include additional the symptoms of PVS. Please see page 11, section 6.

Clinical Presentation of PVS in Premature Infants

The pathophysiologic consequences of PVS are largely determined by the number and severity of stenosed vessels [69]. As such, there is wide range of clinical presentations of premature infants with PVS, including persistent and frequent hypoxemia and respiratory distress, prolonged supplemental oxygen need, inability to wean from respiratory support, and/or new onset PH associated with chronic lung disease [70]. The initial signs and symptoms are often non-specific and are also encountered in preterm infants with PH and BPD alone. In addition to tachypnea, increase work of breathing, and retractions, preterm infants with PVS may also experience new or worsened PH, failure to maintain or gain weight, or require unexplained increases in ventilatory or oxygen support beyond the expected clinical trajectory [3].

Comment 4The authors should describe more specific protocols for transthoracic echocardiography, CT angiography, and cardiac catheterization to improve the detection of PVS.

Response: Thank you for this comment. Currently, there is no unifying evidence-based national or international protocol for detection with variation existing amongst centers. Clearly defined diagnostic criteria and more data in various gestation age is needed (as highlighted by reviewer 2).

We have added language in the section: Challenge of PVS Detection in the NICU: Algorithm for detection PVS in Premature Infants.

Although diagnostic evidence-based guidelines do yet exist for preterm infants with BPD, most experts agree there is a need for developing robust protocols specifically aimed at recognition of risk factors, awareness of clinical signs and symptoms, and utilization of echocardiography as a screening tool for identification of PVS. In high-risk patients or those identified by echocardiography, further evaluation with CT scans, lung perfusion scans, or cardiac catheterization should be considered/performed as currently done at some centers across the country [81]. Finally, consultation with care providers and institutions that have expertise in caring for children with PVS is highly recommended to develop an individualized treatment and follow-up plan.

We have added additional language in the Conclusion and Future Research section to reflect this important point.

With increased recognition of the lack of specific diagnostic protocols and center variation, future efforts must clearly define the diagnostic criteria needed to improve early detection and guide management decisions in the context of various gestation ages.

Reviewer 2

We thank Reveiwer two for helpful comments

The article entitled “Prematurity and pulmonary vein stenosis: The role of parenchymal lung disease and pulmonary vascular disease” has been reviewed. The article was well written and showed detail description and discussion of the development of PVS and the relation of BPD and PVS. It is a new concept and require further investigation.

Comment 1: I would suggest that the authors provide more detail definition of pulmonary vein stenosis, and the correlation between PVS and gestation age of prematurities.

Response 1: Thank you for these comments. We have provided a more detailed definition of PVS. We agree that defined diagnostic criteria and more data in various gestation age would be very helpful in the future research – and as such have added this to our conclusion and future research section.

Introduction. First paragraph: Pulmonary vein stenosis (PVS) is a heterogeneous disease process that contributes to morbidity and mortality in infants and young children [1,2]. PVS describes the pathologic process of intraluminal narrowing of the veins that carry oxygenated blood from the lungs back to the left side of the heart [3].

Comment 2: Clearly defined diagnostic criteria and more data in various gestation age would be very helpful in the future research.

Response 2: We have provided language below to reflect this comments. Please comment 4 from Reviewer 1

Conclusion and Future Research. Last Paragraph: With increased recognition of the lack of specific diagnostic protocols and center-variation, future efforts must clearly define the diagnostic criteria needed to improve early detection and guide management decisions in the context of various gestation ages.

Round 2

Reviewer 1 Report

The authors have revised well according to our comments.